# SYNTHCLIP: ARE WE READY FOR A FULLY SYNTHETIC CLIP TRAINING?

## ABSTRACT

We present SynthCLIP, a CLIP model trained on entirely synthetic text-image pairs. Leveraging recent text-to-image (TTI) networks and large language models (LLM), we generate synthetic datasets of images and corresponding captions at scale, with no human intervention. In this work, we provide an analysis on CLIP models trained on synthetic data. We provide insights on the data generation strategy, number of samples required, scaling trends, and resulting properties. We also introduce SynthCI-30M, a purely synthetic dataset comprising 30 million captioned images. Our work focuses on showing the advantages and disadvantages of synthetic data for training CLIP models. Our code, trained models, and data, will be released as open source.

## 1 INTRODUCTION

Self-supervised training strategies (He et al., 2022; Caron et al., 2021; Chen et al., 2020) are fundamental for many recently released foundation models. These techniques make use of a vast amount of data without incurring a large annotation cost. In particular, contrastive representation learning (Schroff et al., 2015) has been successfully employed to extract joint embeddings for heterogeneous data modalities. By using multi-modal training data, CLIP (Radford et al., 2021b) provides a common representation that links visual and linguistic information. Today, CLIP encoders are included in a wide range of applications, spanning from zero-shot image understanding (Liu et al., 2023b; Ren et al., 2024), to style transfer (Kwon & Ye, 2022), and robotics control (Shridhar et al., 2022), among others.

Training CLIP requires large-scale captioned image datasets that are often collected from the web. Unfortunately, retrieving these data from the internet may present significant disadvantages (Li et al., 2023a; Piktus et al., 2021; Kang et al., 2023). While it is easy to scale the number of unique samples, this comes at an increased difficulty in controlling their quality, which may result in a poor alignment between images and corresponding captions. Moreover, it is challenging to monitor the *content* of the collected images, resulting in potential concerns for illegal[1] or copyrighted content. As an alternative to web-crawled data, some have explored the usage of synthetic data to train supervised (Sariyildiz et al., 2023; He et al., 2023) or self-supervised (Tian et al., 2023) networks.

In this paper, our goal is to investigate the performance of CLIP models trained on fully synthetic data in the form of captioned images, to define practices and properties of synthetic data for training. To achieve our objective, we train SynthCLIP, a CLIP model trained exclusively on large-scale generated data. We propose a pipeline that jointly leverages existing text-to-image (TTI) and large language models (LLM) to produce text-image pairs. The captioned images are generated in an end-to-end fashion starting from a large list of concepts which guarantees variability of the synthesized data. We use the LLM to produce captions starting from sampled concepts, and then synthesize their corresponding images using TTI models. Our pipeline brings a significant advantage: we can generate data at any scale, arbitrarily increasing the size of training data depending only on computational power, *with no human intervention*. This allows for a controlled study on SynthCLIP performance and properties. Our contributions are threefold:

1. We propose SynthCLIP, a CLIP model trained entirely on synthetic data generated with an automatic pipeline that is scalable to any desired dataset size.

---

[1]We report a recent article in mainstream news on the topic.

2. We provide an extensive study on SynthCLIP, including performance evaluation on five tasks and multiple datasets, and a comprehensive analysis of the resulting properties from training on synthetic data.
3. We release SynCI-30M, an entirely synthetic dataset produced using our generation pipeline. It is composed of 30 million pairs of images and corresponding captions. We also release models trained on different synthetic dataset scales, and the code to generate the dataset.

## 2 RELATED WORK

**Representation Learning.** Early works in representation learning on images used pre-text tasks such as inpainting, jigsaw puzzle solving, and image rotation prediction (Pathak et al., 2016; Noroozi & Favaro, 2016; Gidaris et al., 2018). Instead, SimCLR (Chen et al., 2020) leverages contrastive learning to maximize the similarity between two augmented views of the same image. Alternatively, masked autoencoders (MAE) (He et al., 2022) use a masked patch prediction task to learn visual representations. CLIP (Radford et al., 2021a) and other similar works (Mu et al., 2022; Zhai et al., 2023; Fini et al., 2023) use contrastive learning to learn joint visual and textual representations. Language-image pre-training necessitates high-quality text-image pairs. Its core idea is to maximize the similarity between encoded textual and image representation. We study the possibility of generating synthetic text-image pairs for training CLIP-like models starting from simple concepts only.

**Synthetic Captions.** Recent works emphasize the importance of high-quality and aligned text-image pairs when training CLIP models, and propose a synthetic caption generation pipeline for improving it. VeCLIP (Lai et al., 2023) and CapsFusion (Yu et al., 2023a) propose methods to produce aligned captions. Both start with a captioning model such as BLIP (Li et al., 2022) or LLaVA (Liu et al., 2023a), to obtain a semantically-rich synthetic caption. However, captioning models suffer from over-simplification and lack world knowledge, hence they can be compensated by the usage of an LLM (Lai et al., 2023; Yu et al., 2023a). LaCLIP (Fan et al., 2023) improves the text branch of CLIP by leveraging an LLM to provide multiple rewrites of the same caption to use in contrastive learning. While this improves downstream tasks, it may not reflect the content of the image due to hallucinations (Fan et al., 2023). All these works assume the availability of real data, instead, we introduce a fully synthetic pipeline for data generation, allowing arbitrary scalability and control over generated samples.

**Synthetic Data.** Synthetic data has been used in machine learning fields ranging from audio (Rossenbach et al., 2020) to language (Yang et al., 2020; Li et al., 2023b) and vision (Varol et al., 2017; Jahanian et al., 2022; Zhou et al., 2023). In computer vision, they allow for improving models' performance on several downstream tasks such as semantic segmentation (Richter et al., 2016; Ros et al., 2016; Chen et al., 2019), object detection (Johnson-Roberson et al., 2017), and image classification (Yuan et al., 2024; Shmelkov et al., 2018). Recent works have explored the use of synthetic data from TTI models, to augment training on real data (Azizi et al., 2023; Sariyildiz et al., 2023; He et al., 2023). Yu et al. (2023b) uses a framework to generate synthetic images, increasing the diversity of existing datasets. All these assume knowledge about object classes in the downstream task, and work with images only. Recently, StableRep (Tian et al., 2023) showed that synthetic images generated from Stable Diffusion can be used to train self-supervised methods, however, it still uses real captions, limiting scalability and proper analysis. Parallel works (Fan et al., 2024; Tian et al., 2024) preliminarily investigate training of vision models on synthetic captions and images. However, they are not specific to CLIP, and they include a small set of classes for generation, preventing appropriate analysis. We discuss our positioning relative to these works in the appendix.

## 3 TRAINING SYNTHCLIP

In this section, we present the training procedure for SynthCLIP. We show the pipeline in Figure 1. First, we start by identifying a *concept bank* containing many raw visual concepts, *i.e.* words that can be associated with their corresponding representations in images. This broad definition covers either common objects, items, and animals (*e.g.* "cat"), proper nouns and specific elements (*e.g.* "Eiffel Tower"), and intangible items associated with specific visual characteristics (*e.g.* "love", that is often represented with stylized hearts). An LLM is then prompted to generate captions for all the

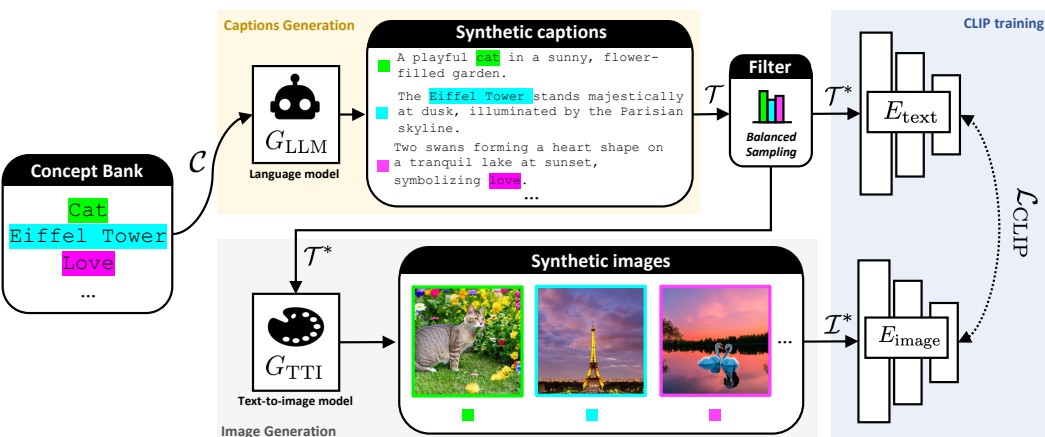

Figure 1: **Pipeline Overview.** From a set of concepts $\mathcal{C}$ (left), we obtain a set of synthetic captions $\mathcal{T}$ with an LLM, further refined to $\mathcal{T}^*$ by a filtering balanced sampling operation (top). Generated captions are used to prompt a text-to-image model, obtaining synthetic images aligned with the caption (bottom). We then train CLIP on the generated text-image pairs (right).

concepts in the concept bank, leading to a set of synthetic captions (Section 3.1). The generated captions are then filtered to a smaller set for improved performance (Section 3.2). These filtered captions are then passed to a text-to-image model to generate corresponding images (Section 3.3). After obtaining our synthetic {caption, image} pairs, a standard CLIP training is carried on the generated data, obtaining the language and text encoders that can be used for downstream tasks (Section 3.4). We now describe each step in detail.

## 3.1 STEP 1: CONCEPT-BASED CAPTIONS GENERATION

The first stage of our pipeline involves the generation of synthetic image captions, that we later aim to use as prompts for text-to-image generators. To achieve this, we utilize an LLM conditioned on our concept bank. The model is prompted to generate captions that describe a scene related to a chosen concept. In our process of generating these captions, we experimented with various prompting techniques, discovering that conditioning the LLM to focus on a particular concept leads to more diverse captions. Concept conditioning ensures that the LLM does not just repeatedly produce captions about a limited set of concepts, over-represented in the LLM training dataset. In other words, this approach helps prevent the model from becoming biased towards certain concepts, and encourages a broader spectrum of caption generation. Limited concept diversity hinders the CLIP training, since contrastive learning highly benefits from variability and more concept coverage (Xu et al., 2023). Hence, diversity is a requirement for a proper analysis of SynthCLIP.

We start by introducing our concept bank $\mathcal{C}$ composed by $N_{\mathcal{C}}$ concepts. Unless otherwise stated, we use the MetaCLIP concept bank (Xu et al., 2023), that contains over 500,000 concepts drawn from WordNet Synsets and Wikipedia common unigrams, bigrams, and titles. We observe that $N_C$ deeply influences CLIP performance, and we investigate this effect in Section 4.3. We then focus on prompt engineering, a critical aspect for generating effective captions for text-to-image generation.

Image generators are sensitive to the quality of the input prompt (Gu et al., 2023), which is often a brief text description capturing the characteristics of the desired image. We set specific requirements to ensure that the prompts generated by the LLM are well-suited for the subsequent image generation:

**(1) Focus on a Single Concept:** Each generated caption should be centered around a single concept, presented in a clear and coherent context.

**(2) Brevity and Clarity:** The prompts need to be concise yet grammatically correct. The goal is to avoid overly complex or vague inputs that could lead to ambiguous or incorrect images.

**(3) Prompt-Only Generation:** Our aim is to have the LLM generate prompts without engaging in

Figure 2: **Generation samples.** We show generated captions and images pairs for concepts "`cat`" and "`Paris`". Our pipeline provides high variability and realistic contextual placement of input concepts.

further reasoning or elaboration. This approach not only saves computational resources, but also simplifies the parsing process.

Assuming $c \in \mathcal{C}$, our designed prompt is:

> Your task is to write me an image caption that includes and visually describes a scene around a concept. Your concept is $c$. Output one single grammatically correct caption that is no longer than 15 words. Do not output any notes, word counts, facts, etc. Output one single sentence only.

Formally, we define our LLM generator as $G_{\text{LLM}}$ and the prompt as $p$. Hence, the set of generated captions is $\mathcal{T} = \{t_{c,n} \sim G_{\text{LLM}}(p, c)\}, \forall c \in \mathcal{C}, \forall n \in \{1, 2, ..., N\}$ where $N$ is the number of desired captions for each concept. By looking at the captions in Figure 2, we show how this mechanism results in a highly variable contextual placement of each concept.

## 3.2 STEP 2: CAPTIONS FILTERING

When generating captions conditioned on a specific concept $c$, it is typical for other concepts $c' \neq c, c' \in \mathcal{C}$ to appear within the same caption. This is expected, since even when a sentence is focused on a single concept, other related concepts emerge within the described scene. For example, if $c =$ "`bird`", a generated caption might be "`a bird is resting on a tree`", introducing an additional concept $c' =$ "`tree`". This LLM-specific behavior may create imbalances in the generated data for CLIP training, which benefits from the usage of a balanced amount of concepts (Xu et al., 2023).

We create a balanced set of captions, $\mathcal{T}^*$, by applying the balancing sampling method proposed in MetaCLIP (Xu et al., 2023) to our setting. It consists of two stages: substring matching, and probability balancing. Substring matching determines which concepts from $\mathcal{C}$ appear in each caption within $\mathcal{T}$. This enables us to measure the real frequency of each described concept across the synthesized captions. Probability balancing is then employed to subsample captions $\mathcal{T}^*$ from $\mathcal{T}$. It increases the probability of selecting captions with long tail concepts, preventing over-representation of frequently-generated concepts. This yields a subset of captions where both frequent and long-tail concepts are adequately represented. Hence, this approach ensures a diverse and task-agnostic captions set suitable for foundation model pre-training. By sizing the parameters of balanced sampling, we are able to choose the size of the subset $\mathcal{T}^*$. For more details, we refer to (Xu et al., 2023).

## 3.3 STEP 3: IMAGE GENERATION

Having successfully created a balanced set of synthetic captions $\mathcal{T}^*$, our next step is to generate the corresponding images. For this, we utilize a text-to-image generator $G_{\text{TTI}}$. We choose Stable Diffusion (Rombach et al., 2022) for this purpose, due to its open-source availability and relatively lower computational demands. For each caption in our set $\mathcal{T}^*$, we generate a corresponding image. This process results in a collection of images, $\mathcal{I}^* = \{x_k \sim G_{\text{TTI}}(t_k)\}$, where each $x_k$ is an image synthesized from the caption $t_k \in \mathcal{T}^*$. In Figure 2, we show how we generate highly aligned images

which correctly capture the described scene and complement it with related realistic information. This proves the efficacy of our caption generation pipeline, leading to appropriate image generation.

### 3.4 STEP 4: CLIP TRAINING

Finally, we use the synthetic text-image pairs to train a CLIP model, to explore how effectively a model can learn from entirely synthetic data. We train two encoders, each one dedicated to either the image or text modality, defined as $E_{\text{image}}$ and $E_{\text{text}}$, respectively. We follow the standard CLIP training pipeline (Radford et al., 2021a), by applying a contrastive loss on the image and text representations through the encoders. Formally, we extract representations $h = E_{\text{image}}(x_k), x_k \in \mathcal{I}^*$ and $z = E_{\text{text}}(t_k), t_k \in \mathcal{T}^*$, and train by minimizing the CLIP loss $\mathcal{L}_{\text{CLIP}}(h, z)$.

## 4 EXPERIMENTS

In this section, we evaluate the performance of SynthCLIP. We start by introducing the experimental setup in Section 4.1, presenting datasets, generation models, and downstream tasks. Section 4.2 benchmarks SynthCLIP against baselines trained on real data on multiple tasks. Finally, Section 4.3 includes an analysis of the properties of SynthCLIP and ablation studies for the introduced components.

### 4.1 EXPERIMENTAL SETUP

**Downstream Tasks** We use five different downstream tasks to assess performance. For ease of evaluation, we categorize the downstream tasks into two categories; **(1) Vision Tasks** and **(2) Vision-Language Tasks**. The first focuses on evaluating the capabilities of the frozen vision encoder $E_{\text{image}}$ only, *i.e.*, linear probing and few-shot classification. The second evaluates the synergy between the image encoder $E_{\text{image}}$ and text encoder $E_{\text{text}}$ together. We use *image retrieval*, *text retrieval*, and *vision-language zero-shot classification tasks* as evaluation tasks, following the original CLIP (Radford et al., 2021a). Since our evaluation pipeline consists of several tasks whose metrics can behave differently, we aggregate performance across all tasks using the $\Delta_{\text{MTL}}$ metric (Vandenhende et al., 2021), where a model with positive $\Delta_{\text{MTL}}$ indicates an overall better performance compared to a reference baseline.

**Datasets** We use the real datasets CC3M (Sharma et al., 2018) ($3 \times 10^6$ samples) and CC12M (Changpinyo et al., 2021) ($8.8 \times 10^6$ samples[2]). Real images come at different resolutions, so we resize the shorter edge of the images to 256px. For SynthCLIP, we generate an entirely synthetic dataset, that we call SynthCI (**Synth**etic **C**aptions-**I**mages) at different scales (number of samples). We refer to SynthCI-3M for a version of SynthCI where $\mathcal{T}^*$ and $\mathcal{I}^*$ include $3 \times 10^6$ captions and images, respectively. For zero-shot evaluation we use ImageNet (Russakovsky et al., 2015), for linear probing and few shot we use CIFAR10 (Krizhevsky et al., 2009a), CIFAR100 (Krizhevsky et al., 2009b), Aircraft (Maji et al., 2013), DTD (Cimpoi et al., 2014), Flowers (Nilsback & Zisserman, 2008), Pets (Parkhi et al., 2012), SUN397 (Xiao et al., 2010), Caltech-101 (Fei-Fei et al., 2004) and Food-101 (Bossard et al., 2014), and for image and text retrieval we use MSCOCO (Lin et al., 2014), Flickr8K (Hodosh et al., 2013) and Flickr30K (Young et al., 2014).

**Caption & Image Generation Models** For caption generation, we use Mistral-7B-Instruct V0.2 (Jiang et al., 2023) with temperature 0.7 and top-p set to 0.95. We also set the presence and frequency penalties at 1. For image synthesis, we use Stable Diffusion v1.5 (Rombach et al., 2022) with classifier-free guidance set to 2 and 50 Denoising Diffusion Implicit Models (DDIM) steps following Tian et al. (2023). The images are generated at $512 \times 512$px and then stored on disk at $256 \times 256$px. It takes 0.9 seconds to generate and save one image on NVIDIA A100 GPU. Image generation was performed on a 48 A100-80GB GPU cluster.

**Training Setup** All trained CLIP models use ViT-B/16 (Dosovitskiy et al., 2021) as $E_{\text{image}}$ and the default CLIP text encoder (Radford et al., 2021a) as $E_{\text{text}}$. $E_{\text{image}}$ and $E_{\text{text}}$ are trained for 40 epochs with a batch size of 4096, a learning rate of $5 \times 10^{-4}$, weight decay of 0.5, cosine scheduler, and

---

[2]The original CC12M is composed of 12M samples. In December 2023, only 8.8M images were available at the linked URLs.

| | Network | Data | Samples ($\times 10^6$) | Synth. data | CIFAR10 | CIFAR100 | Aircraft | DTD | Flowers | Pets | SUN397 | Caltech-101 | Food-101 | Avg |
|---|---|---|---|---|---|---|---|---|---|---|---|---|---|---|
| *Linear Probing* | CLIP | CC3M | 3 | ✗ | 81.8 | 62.7 | 34.7 | 57.3 | 84.1 | 60.5 | 54.3 | 75.6 | 58.7 | 63.3 |
| | | CC12M | 8.8 | ✗ | 91.3 | 73.0 | 48.5 | 69.6 | 92.2 | 81.3 | 68.9 | 88.2 | 77.7 | 76.7 |
| | SynthCLIP | SynthCI-3M | 3 | ✓ | 80.9 | 60.7 | 36.3 | 60.6 | 85.9 | 59.3 | 55.4 | 73.8 | 60.7 | 63.7 |
| | | SynthCI-8.8M | 8.8 | ✓ | 85.9 | 65.9 | 44.0 | 68.7 | 90.0 | 71.8 | 64.2 | 83.0 | 71.6 | 71.7 |
| | | SynthCI-10M | 10 | ✓ | 86.4 | 67.8 | 44.9 | 68.8 | 90.4 | 71.9 | 64.8 | 85.2 | 72.2 | 72.5 |
| | | SynthCI-20M | 20 | ✓ | 87.7 | 68.5 | 47.0 | 70.7 | 92.1 | 75.9 | 68.3 | 86.3 | 75.3 | 74.6 |
| | | SynthCI-30M | 30 | ✓ | 88.0 | 69.6 | 45.3 | 71.0 | 92.4 | 77.6 | 69.0 | 86.2 | 76.0 | 75.0 |
| *Few-shot* | CLIP | CC3M | 3 | ✗ | 61.4 | 70.9 | 45.2 | 73.2 | 93.0 | 71.0 | 93.3 | 91.6 | 68.2 | 74.2 |
| | | CC12M | 8.8 | ✗ | 80.3 | 83.5 | 55.7 | 82.0 | 96.8 | 85.5 | 96.9 | 97.4 | 86.3 | 84.9 |
| | SynthCLIP | SynthCI-3M | 3 | ✓ | 57.6 | 68.8 | 47.2 | 74.3 | 93.5 | 70.8 | 93.5 | 89.9 | 68.3 | 73.8 |
| | | SynthCI-8.8M | 8.8 | ✓ | 62.4 | 73.3 | 56.9 | 79.6 | 95.7 | 80.9 | 95.8 | 95.1 | 78.4 | 79.8 |
| | | SynthCI-10M | 10 | ✓ | 67.0 | 75.1 | 59.3 | 80.4 | 95.9 | 82.8 | 95.9 | 95.4 | 79.4 | 81.2 |
| | | SynthCI-20M | 20 | ✓ | 70.6 | 77.4 | 64.4 | 81.4 | 96.7 | 84.7 | 96.6 | 96.1 | 82.8 | 83.4 |
| | | SynthCI-30M | 30 | ✓ | 74.0 | 80.8 | 66.1 | 82.5 | 97.2 | 86.2 | 96.8 | 96.5 | 83.6 | 84.9 |

(a) Vision Tasks

| | | | | | Image Retrieval | | | | Text Retrieval | | | | 0-shot |
|---|---|---|---|---|---|---|---|---|---|---|---|---|---|
| Network | Data | Samples ($\times 10^6$) | Synth. data | | MS Coco | Flickr 8K | Flickr 30K | Avg | MS Coco | Flickr 8K | Flickr 30K | Avg | Imagenet |
| CLIP | CC3M | 3 | ✗ | | 23.6 | 39.9 | 37.7 | 33.7 | 29.7 | 50.8 | 48.1 | 42.9 | 14.9 |
| | CC12M | 8.8 | ✗ | | 43.8 | 66.2 | 66.8 | 58.9 | 57.4 | 80.3 | 77.3 | 71.7 | 33.6 |
| SynthCLIP | SynthCI-3M | 3 | ✓ | | 21.5 | 39.1 | 41.1 | 33.9 | 28.9 | 53.7 | 55.4 | 46.0 | 9.5 |
| | SynthCI-8.8M | 8.8 | ✓ | | 34.9 | 58.0 | 61.5 | 51.5 | 48.6 | 76.0 | 79.3 | 68.0 | 18.5 |
| | SynthCI-10M | 10 | ✓ | | 36.7 | 58.0 | 64.0 | 52.9 | 50.0 | 75.1 | 81.8 | 69.0 | 20.9 |
| | SynthCI-20M | 20 | ✓ | | 42.5 | 65.4 | 69.2 | 59.0 | 57.8 | 81.7 | 87.5 | 75.7 | 28.0 |
| | SynthCI-30M | 30 | ✓ | | 44.0 | 68.3 | 72.9 | 61.7 | 58.0 | 84.4 | 88.8 | 77.1 | 30.5 |

(b) Vision-Language Tasks

| *SynthCLIP setup* | | *Baseline CLIP* | |
|---|---|---|---|
| Data | Samples ($\times 10^6$) | CC3M | CC12M |
| SynthCI-3M | 3 | -5.60% | -36.0% |
| SynthCI-8.8M | 8.8 | +31.3% | -15.0% |
| SynthCI-10M | 10 | +36.4% | -12.3% |
| SynthCI-20M | 20 | +53.9% | -3.10% |
| SynthCI-30M | 30 | +60.1% | +0.20% |

(c) $\Delta_{\text{MTL}}$ evaluation

Table 1: **Benchmark.** We compare against CLIP models trained on real datasets (CC3M and CC12M). We train SynthCLIP on our synthetic datasets, SynthCI, at various scales. We observe a consistent improvement in performance in both vision (a) and vision-language (b) tasks, as the scale of the SynthCI dataset increase. This demonstrates the scalability advantage of SynthCLIP. In (c) we aggregate multi-task performance with $\Delta_{\text{MTL}}$ across all trained networks.

1 warmup epoch. We use random resized crop with scale $0.5 - 1.0$ as data augmentation. We use the codebase of SLIP (Mu et al., 2022) to train all the models on 16 NVIDIA-V100-32GB GPUs.

## 4.2 BENCHMARK EVALUATION

**Performance on the same data scale** We evaluate the effectiveness of our entirely synthetic data generation pipeline for training CLIP models compared to training on real data. We use CLIP (Radford et al., 2021a) trained on CC3M and CC12M as baselines. We first train SynthCLIP on two versions of SynthCI each matching the data scale of CC3M and CC12M, which we call SynthCI-3M and SynthCI-8.8M, respectively. We report the performance on vision tasks in Table 1a and vision-language tasks in Table 1b, aggregating all metrics with $\Delta_{\text{MTL}}$ (Vandenhende et al., 2021) in Table 1c (details about $\Delta_{\text{MTL}}$ are provided in Supplementary). As visible in Table 1c, we obtain lower performance when both datasets are composed by $3 \times 10^6$ samples (**-5.60%**) and $8.8 \times 10^6$ samples (**-15.0%**), compared to the corresponding real data training with the same dataset size (CC3M and CC12M, respectively). This is expected: considering that real and synthetic data differ in distribution, while training on synthetic samples and testing on real ones, we incur a distribution shift, which ultimately harms performance (Zhou et al., 2023; Fan et al., 2024).

**Scaling SynthCLIP** Our objective now is to analyze the scaling of SynthCLIP, to match performance obtainable by training CLIP on real data. It is indeed known that bigger training datasets help to increase performance (Radford et al., 2021b). However, while scaling real datasets necessitates custom collection pipelines from different sources and data curation, we exploit the advantage

Figure 3: **Performance improvement for different SynthCI scales.** We show the improvements for all metrics with respect to SynthCLIP trained on SynthCI-3M. Vision-language tasks exhibit better absolute improvements and less saturation with respect to vision ones.

| Network | SynthCI samples $(\times 10^6)$ | CC12M samples $(\times 10^6)$ | Lin. Prob. | Few-shot | Img Ret. | Text Ret. | IN 0-shot | $\Delta_{\text{MTL}}$ |
|---|---|---|---|---|---|---|---|---|
| CLIP | - | 8.8 | 76.7 | 84.9 | 58.9 | 71.7 | 33.6 | ↕ |
| SynthCLIP | 30 | - | 75.0 | 84.9 | 61.7 | 77.1 | 30.5 | +0.2% |
| Finetuned *(5 epochs)* | 30 | 0.5 | 76.2 | 86.0 | 67.5 | 74.7 | 34.4 | +4.4% |
| | 30 | 2.0 | 76.1 | 86.1 | 67.8 | 79.3 | 35.2 | +6.2% |
| | 30 | 8.8 | **77.3** | **86.9** | **68.6** | **80.8** | 37.9 | +9.0% |
| Finetuned *(10 epochs)* | 30 | 0.5 | 76.0 | 86.0 | 67.2 | 78.7 | 34.5 | +5.4% |
| | 30 | 2.0 | 76.5 | 86.0 | 67.9 | 79.8 | 36.0 | +7.0% |
| | 30 | 8.8 | **77.3** | 86.6 | **68.6** | 80.4 | **38.6** | **+9.3%** |

Table 2: **Finetuning on real data.** We finetune SynthCLIP using different amounts of data from CC12M and epochs. We significantly improve performance compared to both real-only or synthetic-only setups, proving that synthetic trainings may serve as a strong initialization for vision-language representation learning.

of our data synthesis pipeline, *i.e.* the capability to scale the size of the training data with no human intervention. In practice, we simply let our generation script run for longer, and re-train SynthCLIP on the larger SynthCI version obtained doing so. In particular, we report performance for SynthCLIP trained on $\{10 \times 10^6, 20 \times 10^6, 30 \times 10^6\}$ SynthCI samples, finally matching with 30 million samples the performance of the biggest model we trained on real data (CLIP on CC12M), against which we achieve $\Delta_{\text{MTL}} = $ **+0.20%**. We also report a significant increase with respect to CLIP trained on CC3M ($\Delta_{\text{MTL}} = $ **+60.1%**). From a single task perspective, we outperform CLIP trained on CC12M on image and text retrieval (**+2.8%** and **+5.4%**, respectively), while performing competitively with linear probing (**-1.7%**) and few-shot (**+0.00%**). While we still underperform in zero-shot evaluation (**-3.1%**), we attribute this also to additional bias effects that we study in Section 4.3. Ultimately, our experiment shows that by conditioning on a generic list of visual concepts, SynthCLIP can scale and match CLIP trained on large datasets, albeit using more samples due to the distribution shift between our synthetic training data and real test datasets.

**Scaling trends** To ease understanding to which extent scaling training data influences each task, we plot percentage improvements for each task in Figure 3, assuming as reference the performance achieved with SynthCLIP trained on SynthCI-3M. As visible from the plot, vision-language tasks (green, red, purple curves) tend to achieve more significant performance increase with respect to vision (blue, orange). We attribute this to the good quality of our captions. Our two-step generation pipeline produces text-image pairs that are always fairly aligned with the corresponding image. This synthetic data property further mitigates the distribution shift at scale.

**SynthCLIP as pre-training** While synthetic data allow scalability, the effects of the distribution shift are still significant. We now investigate if these could be mitigated by using real data. It is well-known how training on mixed datasets of real and synthetic samples can boost performance (Fan et al., 2024; Yuan et al., 2024), as we also report in the appendix. We instead propose a different setup, designed to highlight the advantages of fully synthetic trainings. We introduce a *Finetuning* scenario, in which we first pre-train SynthCLIP on SynthCI-30M, and we subsequently finetune it, with the same contrastive loss, on CC12M, for a limited number of epochs. We assume no access to previously used synthetic samples. This reflects real-world practices in which foundation models are first pre-trained, and further adapted to the users' needs (Caron et al., 2021; Oquab et al., 2024). We report results in Table 2. Surprisingly, we show a significant improvement in performance, for all settings. Increasing the number of epochs leads to better performance on real datasets, with the Finetuned model on full CC12M for 10 epochs performing best (**+9.3%**). However, even 5 epochs on $1.0 \times 10^6$ real samples bring considerable advantages, outperforming the CLIP training on the full CC12M by a **+4.4%** $\Delta_{\text{MTL}}$ margin. This happens thanks to the strong features extracted by SynthCLIP, resulting from training on aligned and varied captioned images. Ultimately, we show that we can effectively compensate for the distribution shift with small real datasets.

| Network | Synth. Images | Synth. Captions | Linear Probing | Few-shot | Img retrieval | Text retrieval | IN 0-shot |
|---|---|---|---|---|---|---|---|
| CLIP | ✗ | ✗ | 63.6 | 74.2 | 33.7 | 42.9 | 14.9 |
| CLIP + Text-to-image | ✓ | ✗ | 65.1 | 74.2 | 41.2 | 51.7 | **15.4** |
| CLIP + Captioning | ✗ | ✓ | **70.1** | **78.1** | **46.0** | **62.4** | 12.4 |
| SynthCLIP | ✓ | ✓ | 63.7 | 73.8 | 33.9 | 46.0 | 9.5 |
| SynthCLIP + Captioning | ✓ | ✓ | 66.5 | 74.3 | 43.5 | 57.1 | 8.5 |

(a) Quantitative evaluation          (b) Captioning examples on SynthCI data

Figure 4: **Which synthetic data modality matters more?** We assess which synthetic modality impacts performance more by combining real/synthetic captions/images (a). The usage of real captions implies using the original captions from CC3M. "Synthetic captions" refers to either captions generated by LLaVA (Liu et al., 2023a) ("Captioning") or an LLM (SynthCLIP). "Synthetic images" refers to generated images from Stable Diffusion. Prompts can be either real (CLIP + Text-to-image) or synthetic (SynthCLIP). Captioning with LLAVA improves performance even in SynthCLIP, due to corrections (b), where elements in the prompt missing in generated images are underlined in red.

## 4.3 ANALYSIS

Here, we conduct an analysis to examine key aspects of SynthCLIP. We focus on the importance of textual and visual data modalities, ablate pipeline components (data filtering technique and LLM used for captions generation), and quantify the effects of $\mathcal{C}$ on performance. For all tests, we train on 3 million samples, *i.e.*, a similar scale to CC3M, due to the high computational cost of the larger experiments.

**Do synthetic captions or synthetic images matter more?** SynthCLIP uses entirely synthetic text-image pairs. A key question arises: which has a greater impact on the model's performance in downstream tasks – synthetic images or synthetic captions? In Table 4a, we compare the standard CLIP model trained on CC3M, SynthCLIP, and two hybrid CLIP variants. One hybrid uses real captions with synthetic images (CLIP + Text-to-Image), generated using Stable Diffusion v1.5, while the other pairs real images with synthetic captions (CLIP + Captioning), created with the LLaVA (Liu et al., 2023a) model. Note that these, requiring one real modality, are less scalable than SynthCLIP.

We measure that CLIP + Captioning significantly outperforms standard CLIP in several benchmarks, indicating the effectiveness of synthetic captions in CLIP training. For instance, this approach improves linear probing by 6.5% and text retrieval by 19.5%, though it slightly decreases zero-shot performance by 2.5%. On the other hand, CLIP + Text-to-Image shows less marked improvements and no gains in few-shot performance. This suggests that keeping images real and recaptioning them is more advantageous than generating images for real captions, possibly due to domain shifts and content generation mismatches in synthetic images as noted in Gani et al. (2023); Wu et al. (2023).

Hence, we introduce SynthCLIP+Captioning as an extra baseline. Given that TTI models could miss details in text prompts, recaptioning generated images can be beneficial. This is evident in Figure 4b, where recaptioning corrects alignment issues from the image generation process (*e.g.* the missing `bench` in the generated image). Comparing SynthCLIP and SynthCLIP+Captioning in Table 4a (rows 4 and 5) shows significant gains with captioning, such as a 9.6% improvement in image retrieval. These results advocate for a potential boost in performance from future developments in prompt faithfulness of the TTI model. Moreover, it also opens doors to combinations of caption enrichment techniques such as VeCLIP (Lai et al., 2023) and CapsFusion (Yu et al., 2023a).

**Data Filtering Ablation** In creating our SynthCI-$X$ datasets in Section 4.2, we utilized balanced sampling to select a desired number of captions from a larger set of generated ones. In this section we want to assess how different data sampling strategies affect SynthCLIP's performance. We focus on the impact of substituting balanced sampling with a more straightforward *random sampling* approach. For this, we randomly choose a subset of $3 \times 10^6$ captions from $\mathcal{T}$. The corresponding images for these randomly selected captions are generated using Stable Diffusion v1.5, following the same procedure presented in Section 4.1. We then train SynthCLIP on this newly formed dataset. The results, presented in Table 3a, indicate a noticeable decline in performance across various tasks with random sampling, especially in retrieval tasks. Here, we observe a drop of 2.7% in both image

| Network | Lin. Prob. | Few-shot | Img Ret. | Text Ret. | IN 0-shot |
|---|---|---|---|---|---|
| SynthCLIP | **63.7** | **73.8** | **33.9** | **46.0** | **9.5** |
| ↳ w/ rand. sampling | 61.5 (-2.2) | 72.0 (-1.8) | 31.2 (-2.7) | 43.3 (-2.7) | 9.4 (-0.1) |

(a) Balanced Sampling vs Random Sampling

| LLM | Lin. Prob. | Few-shot | Img Ret. | Text Ret. | IN 0-shot |
|---|---|---|---|---|---|
| Mistral 7B | **63.7** | **73.8** | **33.9** | **46.0** | **9.5** |
| Vicuna 33B | 61.4 (-2.3) | 69.4 (-4.4) | 26.1 (-7.8) | 36.5 (-9.5) | 8.2 (-1.3) |

(b) Results with a different LLM for captions

| Backbone | Lin. Prob. | Few-shot | Img Ret. | Text Ret. | IN 0-shot |
|---|---|---|---|---|---|
| ViT-B | **63.7** | **73.8** | **33.9** | **46.0** | **9.5** |
| ViT-S | 59.5 (-4.2) | 70.3 (-3.5) | 29.2 (-4.7) | 39.7 (-6.3) | 8.1 (-1.4) |

(c) Results with a different backbone.

Table 3: **Training Components Ablations.** Table (a) proves the effectiveness of balanced sampling. Table (b) compares two LLMs, showing Mistral-7B's consistent advantage across various tasks. Table (c) tests a smaller backbone model, proving the expected scaling.

| Concepts | $N_c$ $(\times 10^3)$ | Lin. Prob. | Few-shot | Img Ret. | Text Ret. | IN 0-shot |
|---|---|---|---|---|---|---|
| $\mathcal{C}$ | 500 | 63.7 | 73.8 | 33.9 | 46.0 | 9.5 |
| $\mathcal{C}_{\text{CC3M}}$ | 40 | **65.4** (+1.7) | **74.8** (+1.0) | **37.1** (+3.2) | **49.9** (+3.9) | **12.6** (+3.1) |
| $\mathcal{C}_{\text{rand}}$ | 40 | 63.1 (-0.6) | 72.9 (-0.9) | 31.8 (-2.1) | 44.8 (-1.2) | 9.2 (-0.3) |

Table 4: **Effect of Concept Bank Size.** We compare SynthCLIP model performance using different concept bank sizes: the full $500 \times 10^3$ concepts ($\mathcal{C}$), a $40 \times 10^3$ subset from CC3M ($\mathcal{C}_{\text{CC3M}}$), and a randomly selected $40 \times 10^3$ subset ($\mathcal{C}_{\text{rand}}$), with each trained on 3 million samples. Results show that models trained on CC3M-specific concepts outperform those using the full concept list or a random selection, when a limited number of samples is used. This suggests a distribution bias in CC3M. Hence, in real deployments, this shows the interest of extending $\mathcal{C}$ as much as possible, proving the interest of using our large concept bank.

and text retrieval compared to balanced sampling. These results underline the critical role of balanced concept distribution for CLIP training, highlighting an advantage of synthetic data for data curation.

**Evaluating Different Language Models for Caption Generation** In Table 3b, we study the effect of changing the language model from Mistral V0.2 7B model to Vicuna 33B. We find that using Mistral V0.2 7B consistently achieves better performance when compared to Vicuna 33B. This might be attributed to Mistral's superior performance on instruction-following benchmarks such as AlpacaEval (Li et al., 2023c). Indeed, we phrase caption generation as an instruction-following task as previously described in Section 3.1. This suggests that with increasingly performing models in instruction following, it will be possible to further improve performances of SynthCLIP training.

**Model size** In order to study the effect of model scaling, we variate the size of the trained model. We train SynthCLIP replacing the ViT-B used for the rest of the experiments with a ViT-S, on 3 million samples from SynthCI. We report average results in all tasks in Table 3c. Consistently with expected results on different model sizes (Radford et al., 2021b), we notice that bigger backbones guarantee better performance in all tasks. This proves the quality of our generated data. Indeed, the training is not saturated in our setup, proving that scaling is not limited by the data size in the setup under analysis.

**Concept Bank Impact** We explore how the concept bank size $\mathcal{C}$ and the type of concepts it contains affect the downstream performance of the model. For this, we create two subsets of $\mathcal{C}$. The first subset, $\mathcal{C}_{\text{CC3M}}$, is derived by identifying the concepts that appear in CC3M captions, by performing substring matching with concepts included in $\mathcal{C}$. This results in $40 \times 10^3$ CC3M-related concepts. The second, $\mathcal{C}_{\text{rand}}$, is formed by randomly selecting the same number of concepts as in $\mathcal{C}_{\text{CC3M}}$ from $\mathcal{C}$.

We generate 3M images for each of $\mathcal{C}_{\text{CC3M}}$ and $\mathcal{C}_{\text{rand}}$ and train SynthCLIP on the generated datasets. We show results in Table 4. Interestingly, we noticed that focusing on CC3M-specific concepts ($\mathcal{C}_{\text{CC3M}}$) enhances performance compared to training with the full $\mathcal{C}$. For example, using $\mathcal{C}_{\text{CC3M}}$ yields a 3.9% improvement in text retrieval and 1.6% in linear probing. We hypothesize that this might be because $\mathcal{C}_{\text{CC3M}}$'s concepts are more aligned with concepts appearing in the downstream tasks, hence indicating a potential distribution bias in CC3M towards concepts prevalent in downstream task images. In contrast, using $\mathcal{C}_{\text{rand}}$ leads to lower performance in all tasks compared to the full $\mathcal{C}$. For example, we observe a 1.2% decrease in text retrieval and 0.8% in linear probing, likely because $\mathcal{C}_{\text{rand}}$'s concepts are less relevant to the downstream tasks. Hence, when specific insights about downstream tasks are unavailable, it is preferable to train on the widest possible range of concepts.

## 5 DISCUSSION

Here, we discuss additional consequential properties of SynthCLIP, resulting from the usage of synthetic data, and we draw conclusions.

**Mitigation of long-tail effects.** While the distribution shift is preventing matching performance on the same data scale, we argue that the control over $\mathcal{C}$ may be used to mitigate long-tail effects. We provide preliminary insights by evaluating zero-shot classification accuracy on 10 classes (listed in appendix) included in $\mathcal{C}$, but undetected in CC12M with substring matching. Importantly, we compare both SynthCLIP and CLIP trained on $8.8 \times 10^6$ images. For evaluation, we collect 150 samples per class. We report 44.18/**60.04** accuracy for CLIP/SynthCLIP. This gives preliminary insights into how synthetic data could be used for training CLIP models resistant to long-tail distributions.

**Data safety** Another aspect resulting from the control of $\mathcal{C}$ is the potential for training SynthCLIP models exclusively on safe data. This is not always possible with web-collected data (Schuhmann et al., 2022), due to image-based NSFW detectors having limited performance (Schramowski et al., 2022), and for the subjective nature of offensive content. We argue that our concept-based caption generation may help the synthesis of safe images only. In $\mathcal{C}$, we did not modify the concepts proposed by Xu et al. (2023) for reproducibility. However, as a preliminary insight, we process $\mathcal{C}$ with an LLM (more details in appendix), detecting 3.15% NSFW concepts that can be filtered from the original $\mathcal{C}$. Moreover, our data synthesis pipeline could be associated with recent approaches for safe image generation (Schramowski et al., 2023). This opens possibilities for safe CLIP trainings.

**Concluding remarks** We presented SynthCLIP, a CLIP model trained exclusively on synthetic data. Our experiments show SynthCLIP's scalability and capability to match the performance of models trained on real data, given enough samples, on a large concept corpus. This paves new ways for entirely synthetic training at scale, possibly extending the capabilities of CLIP. Our investigation of the properties of SynthCLIP reveals novel insights into the role of synthetic data in vision-language models.

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

## A  COMPARISON WITH CONCURRENT WORKS

We now discuss our positioning with respect to recent concurrent works.

- **Concepts.** In Fan et al. (2024), they propose a study on large-scale training of synthetic data for supervised learning and CLIP paradigms. They restrict the concept bank to ImageNet classes and provide several techniques to prompt text-to-image models. We refer the reader to section 3.1 of Fan et al. (2024) for more details on the different proposed prompting techniques and TTI generation pipeline. In the supervised setting, it is clear from Table-A7 of their supplementary material that the performance of training on synthetic data becomes comparable when there are **8 times** more synthetic data available at classifier free configuration scale of 2. Beyond that, the performance of training on synthetic data starts to saturate. Additionally, in that work, the authors assume some class-priors and restrict their training to ImageNet classes, which has been shown in Hammoud et al. (2024) not to be generalizable to classes that are significantly different than those in ImageNet. Additionally, as mentioned in Section 5, training on a larger concept bank may lead to a better performance on long-tail concepts and hence the importance of covering many concepts during training beyond ImageNet classes. In Tian et al. (2024), the generation is also conditioned on the downstream classes. We refer the reader to Table-11 of their supplementary material. This gives you an unfair advantage over real data that is not necessarily conditioned on these classes (Section 4.3). For StableRep, they require the existence of real captions which they utilize as prompts for TTI model which is not scalable and have a limited effectiveness compared to synthetic higher quality captions (Section 4.3).

- **CLIP-specific tasks.** The main difference in our work is that we focus on CLIP-related tasks in a holistic manner. Indeed, Fan et al. (2024) only proposes zero-shot evaluation, with no reasoning about how different tasks are impacted. Tian et al. (2024) only proposes very limited experiments with linear probing on ImageNet. Instead, we focus on 5 different tasks, and contextualize separately the effects of synthetic data on different tasks.

- **Data generation pipeline.** Fan et al. (2024) uses multiple strategies for captions generation, that do not necessarily work with large concept banks. We also provide an experiment on this in B.3. Tian et al. (2024) generation pipeline is the most similar to ours, but relies on in-context examples generated by GPT-4. In our preliminary experiments, we verify that caption generation with in-context learning on Mistral 7B causes the model to collapse (Section B), generating variations of the in-context samples by just replacing $c$. Hence, our designed prompt is more adaptable to different models, as we show in Table 3b of the main paper.

## B  ADDITIONAL ABLATION STUDIES

### B.1  CONCEPT APPEARANCE

In this section, we examine the presence of concepts from our extensive $500 \times 10^3$ concept bank within real text-image datasets like CC3M and CC12M, as well as our SynthCI synthetic datasets. Our method involves substring matching, where we identify and count the occurrences of each concept within the captions of these datasets. This count reveals how frequently different concepts appear, particularly those occurring more than a specified number of times ($k$).

Table 5 summarizes these findings. Notably, even the smallest SynthCI-3M dataset contains significantly more concepts than the larger real CC12M dataset, surpassing it by nearly 2.5 times in terms of concepts appearing at least once ($k = 1$). This trend of broader concept coverage in SynthCI

| Dataset | Concept Appearance | | | Average Appearance |
|---|---|---|---|---|
| | $k = 1$ | $k = 25$ | $k = 50$ | $k \geq 25$ |
| CC3M | $3.9 \times 10^4$ | $1.8 \times 10^4$ | $1.4 \times 10^4$ | $1.4 \times 10^3$ |
| SynthCI-3M | $3.0 \times 10^5$ | $3.6 \times 10^4$ | $2.3 \times 10^4$ | $1.0 \times 10^3$ |
| CC12M | $1.3 \times 10^5$ | $4.8 \times 10^4$ | $3.7 \times 10^4$ | $2.0 \times 10^3$ |
| SynthCI-8.8 | $3.4 \times 10^5$ | $2.3 \times 10^5$ | $5.6 \times 10^4$ | $6.2 \times 10^2$ |
| SynthCI-10M | $3.4 \times 10^5$ | $2.3 \times 10^5$ | $8.5 \times 10^4$ | $7.0 \times 10^2$ |
| SynthCI-20M | $3.4 \times 10^5$ | $2.3 \times 10^5$ | $1.9 \times 10^5$ | $1.4 \times 10^3$ |
| SynthCI-30M | $3.5 \times 10^5$ | $2.4 \times 10^5$ | $1.9 \times 10^5$ | $2.0 \times 10^3$ |

Table 5: **Concept Appearance in Real vs. Synthetic Datasets.** We compares the frequency of concept appearances in real datasets (CC3M, CC12M) and their synthetic counterparts. We report the number of concepts that appear at least $k$ times, along with the average appearances for concepts occurring at least 25 times.

| Data ($\times 10^6$) | Samples ($\times 10^6$) | Backbone | Lin. Prob. | Few-shot | Img Ret. | Text Ret. | IN 0-shot |
|---|---|---|---|---|---|---|---|
| SynthCI-3.3M | 3.3 | ViT-B | **63.7** | **73.8** | **33.9** | **46.0** | **9.5** |
| | | ViT-S | 59.5 | 70.3 | 29.2 | 39.7 | 8.1 |
| SynthCI-10M | 10 | ViT-B | **72.5** | **81.2** | **52.9** | **69.0** | **20.9** |
| | | ViT-S | 69.1 | 79.4 | 50.9 | 66.4 | 18.1 |

Table 6: **Model size ablation.** Bigger models such as ViT-B benefit more from the provided data, as expected. This proves the correctness of our data synthesis procedure.

datasets persists even when increasing the threshold to $k = 25$ or $k = 50$. An intriguing aspect is the average number of samples per concept. The last column of Table 5 shows the average frequency of concept occurrences, considering only those appearing at least 25 times. While CC3M and CC12M, with fewer overall concepts, exhibit a higher average of samples per concept, our SynthCI datasets generally show lower averages. However, SynthCI-30M shows the same average as real datasets, particularly CC12M. This similarity in samples per concept at 30M scale could be a key factor in SynthCI-30M matching the performance of CC12M.

### B.2 MODEL SIZE

In order to study further the effect of model scaling, we extend the training of the smaller transformer ViT-S using Synth-10M. We compare with the results reported in the main paper (Table 3c), while reporting new average results in all tasks in Table 6. We noticed that the SynthCLIP training greatly benefits from further data variability, while performances gap between different transformers sizes are not significantly changed. This further advocates for the stability of the training resulting from our data synthesis pipeline.

### B.3 LLM USAGE

We ablate the importance of the LLM by removing it completely from our generation pipeline and replacing that with CLIP templates for zero-shot classification (Radford et al., 2021b). The rest of our pipeline is unchanged. We generate 3 million samples and train a CLIP model on top of this newly generated set. We call this model NaiveCLIP. We evaluated vision-language tasks, in particular zero-shot classification (ZS), text retrieval (TR), and image retrieval (IR), as presented in Section 4.1 of the main paper. We report for NaiveCLIP/SynthCLIP 2.0/**9.1** for ZS, 2.9/**33.9** for IR, 5.1/**46.0** for TR. From our results, it is evident how the presence of the LLM is necessary to grant the best performance for vision-language tasks.

### B.4 PROMPT ENGINEERING

In this section we showcase alternative attempts to generate synthetic captions.

**Attempt 1 - Generate Captions without Any Conditioning** In our first attempt, we tried to let LLM generate any topic without any conditioning. This was done using the prompt shown in Figure 5. Unfortunately, the captions were overly descriptive and hard for the text-to-image model to generate images for. Moreover, they were always focused on nature, resulting in low variability unsuitable for CLIP training. Examples of generated captions are:

- `A sunlit garden: vibrant roses bloom against a brick wall, butterflies dance around, water droplets sparkle on leaves, soft focus, balanced composition.`

You are an expert image descriptions generator. Your task is to write an image caption to describe a scene that can be used with text-to-image generation model such as DALL-E.
Your description should vividly and descriptively detail the scene to guide the image generator in producing its visualizations of the caption depiction. Use simple words, and the rewrite has to be less than 15 words.
Use modifiers such as lighting, focus, balance, composition, angle, reflections, textures, color palette, style, tone, effects, lens type, mood, artist or photographer name, and more.

Figure 5: **Attempt 1 - Captions Generation Prompt**

Space, Celestial Bodies, Nature, Natural Landscapes, Plants, Trees, Flowers, Domestic Animals, Wild Animals, Gadgets and Electronics, Historical Landmarks and Monuments, Oceans, Marine Life, Underwater Scenery, Mountains, Geographical Features, Urban Landscapes, Cityscapes, Art, Sculptures, Visual Arts, Festivals ,Cultural Events, Celebrations, Vehicles and Transportation, Sports ,Recreational Activities, Architecture,Buildings, Fashion, Clothing, Accessories, Food, Cuisine, Culinary Arts, Weather and Atmospheric Phenomena, Astronomy ,Astrophysics, Musical Instruments, Performances, Traditional and Folk Crafts, Books, Literature, Written Works, Films, Movies, Theater, Dance and Performing Arts, Educational and Scientific Concepts, Health, Medicine, Wellness, Fantasy, Mythology, and Folklore, Video Games and Virtual Worlds, Historical Eras and Civilizations, Celebrities,Public Figures, Influencers, Insects, Microscopic Life, Small Creatures, Tools, Machinery, Industrial Equipment, Toys, Games, Children's Entertainment, Work Environments, Professions, and Occupations, Religious, Spiritual, and Mystical Symbols, Political, Social, and Environmental Movements, Everyday Household Objects and Utilities, Landscapes of Other Planets and Moons, Dinosaurs, Prehistoric Life, Paleo-Scenery, Kitchen Utensils, Cooking Tools, Home Accessories, Interior Decor, Office Furniture, Home Furniture, Gardens, Horticulture, Landscaping, Pets and Companion Animals, Aquatic and Water-based Activities, Educational and Learning Materials, Traditional Clothing, Ethnic Clothing, Body Art and Tattoos, Streets Roads, and Highways, Forests, Jungles, and Wilderness Areas, Planes, Boats, Photography, Robots, Futuristic

Figure 6: **Attempt 2 - Topics Bank**

```
• Sunset over tranquil lake:  A solitary kayaker paddles
  through golden reflection, mountains in distance bathed in
  warm light.  Focus on kayaker's determined face, balanced
  composition.  Soft toned, impressionistic brushstrokes.
• A sunlit garden:  vibrant roses bloom against a weathered
  brick wall, butterflies dance around ripe strawberries on
  a red table, children play nearby, laughter echoes softly.
  Warmth radiates from every detail.
```

**Attempt 2 - Generate Captions Using a Topics Bank**    Instead of having a concept bank that we generate synthetic captions for, our first attempt was to try having a broader list of topics, *i.e.* a topic bank, used for conditional generation. Particularly, we used the topics shown in Figure 6 and then used the prompt shown in Figure 7 to generate the captions.

The observed issue is that for each topic the LLM had some kind of favourable instance. For example for "Wild Animals", most generated captions were about leopards:

```
• In the desert, a leopard is dragging its kill.
• A leopard carries its prey through the arid desert
  landscape.
```

> Image captions are usually composed of four components: (1) Subject: This is the main focus of the sentence. (2) Action or State: This describes what the subject is doing or the state they are in. (3) Setting or Context: This provides additional information about where the action is taking place or the context surrounding the subject. (4) Additional Descriptors: These are adjectives or additional details that provide more depth or description to the subject or setting.
> Your task is to write image captions as described above. You are provided with a "Concept List" below which contains categories you can write captions for.
> Concept List: {sampled_topics_string}
> Rules:
> (1) You are allowed a maximum of 15 words per image caption.
> (2) Select at random a subject from the concept list or be creative and go beyond the list.
> (3) Select a highly specific subject from the concept list for your caption. Avoid general categories; instead, choose a detailed and particular item, creature, or concept. The chosen subject should be a distinct and unique example within its broader category, reflecting your creativity and precision. This means you will provide names, breeds, locations, brands ... all of which ensure specificity.
> (4) Write down the captions without specifying what category it belongs to.
> Rule (3) is very important.
> Please provide me with 10 captions following all the rules above.

Figure 7: **Attempt 2 - Captions Generation Prompt**

- `The majestic snow leopard roams high within Himalayas mountain range territory.`

This issue was not resolvable by adjusting the prompt or parameters of the LLM including the seed, temperature and top-p value. Interestingly, this signals that biases in concept-oriented generations in LLM are significant regardless the amount of data they are trained on. Since we were mostly interested in maximizing the variability of generated concepts, we opted for the concept-based generation pipeline presented in Section 3.1.

**Attempt 3 - Using in-context learning** Additionally we tried prompting the language model by providing in-context sample prompts to guide the generation. In our tested settings, this causes the model to repeat often the in-context prompt, while modifying the concept $c$. For example, if an in-context sample about $c = $ `` `cat'' `` is provided as "`A beautiful cat sitting near Eiffel tower`" and the target concept $c = $ `` `dog'' ``, the generated caption is "`A beautiful dog sitting near Eiffel tower`". Instead, our engineered prompt allows us to generate variable samples by only conditioning on $c$.

## C  DETAILS ON DISCUSSION

**Long-tail effects** In Section 5, we highlight the increased performance on long-tail distributions. The concepts we use for the evaluation are: "`amber`" "`chicks`" "`crystal`" "`crystal_ball`" "`loincloth`" "`rose`" "`sandcastles`" "`sundress`" "`veggies`" "`x-ray`". Although we outperform CLIP considerably, we still report that 44.18 in zero-shot classification is remarkable for unseen classes. We attribute this behavior to the presence of similar concepts in CC12 (*e.g.* "`dress`").

**Safety** As anticipated in Section 5, we performed preliminary safety analysis on the concepts in $\mathcal{C}$. We use LLaMA-3 (Meta, 2024) to process the input concepts with the following prompt:

| Network | CC12M samples $(\times 10^6)$ | SynthCI samples $(\times 10^6)$ | Lin. Prob. | Few-shot | Img Ret. | Text Ret. | IN 0-shot |
|---|---|---|---|---|---|---|---|
| CLIP | 8.8 | - | 76.7 | 84.9 | 58.9 | 71.7 | 33.6 |
| SynthCLIP | - | 20 | 74.6 | 83.4 | 59.0 | 75.7 | 28.0 |
| | - | 30 | 75.0 | 84.9 | 61.7 | 77.1 | 30.5 |
| *Mixed* | 8.8 | 11.2 | **78.7** | **87.0** | **64.7** | **79.6** | **39.1** |

Table 7: **Quantitative evaluation on mixed real-synthetic data**. *Mixed* is obtained by mixing 8.8M real samples of CC12M and 11.2M synthetic samples from SynthCI-20M. Mixing real data with synthetic not only outperforms the best real checkpoint (CLIP on CC12M) and SynthCLIP on the same data scale, but it also outperforms our best SynthCLIP model.

| Network | Lin. Prob. | Few-shot | Img Ret. | Text Ret. | IN 0-shot |
|---|---|---|---|---|---|
| OpenAI CLIP | 85.7 | 92.4 | 75.6 | 88.1 | **68.6** |
| OpenAI CLIP + SynthCI finetuning | **85.8** | **92.6** | **78.1** | **89.8** | 66.7 |

Table 8: **Finetuning the OpenAI CLIP**. The OpenAI CLIP text encoder is used in Stable Diffusion. To prove that our pipeline is not limited by its performance, we finetune it on SynthCI for a reduced number of steps. This is sufficient to improve performance on almost all tasks, proving that our data creation pipeline allows the creation of novel content.

> You will be my assistant in flagging concepts that might be NSFW meaning they are sexual, harmful, racist, abusive or explicit content. You will be presented with a concept and you will return 1 if the sample is NSFW and 0 otherwise. You will only return either 1 or 0 and nothing else. No explanations or anything are needed. Only flag concepts you are sure about being NSFW.

This results in 3.15% of concepts being flagged. Upon manual inspection, the classification seems reliable. We report selected examples of filtered concepts (Warning: potentially offensive content): "`pornography`", "`naked`", "`drunk`", "`weed`", "`escort`".

## D  TRAINING ON MIXED REAL DATA

While mixed training is beyond the scope of our work, we conduct preliminary experiments to understand whether SynthCI could be used jointly to real data for boosting SynthCLIP performance. To do that, we train on a *Mixed* setup, in which we combine CC12M with $11.2 \times 10^6$ synthetic samples from SynthCI, totaling $20 \times 10^6$ captioned images. Note that this allows to compare a SynthCLIP model trained with the same computational costs, trained on SynthCI-20M. We find that this indeed improves the performance over pre-training on SynthCI-30M and CC12M across all benchmarks as seen in Table 7. This means that our synthetic text-image generation pipeline could be used in conjunction with existing large-scale curated datasets to achieve the best performance, in agreement with the literature (Yuan et al., 2024; He et al., 2023).

## E  DISCUSSION ON STABLE DIFFUSION

Stable Diffusion v1.5 uses the OpenAI CLIP textual encoder for prompt encoding, and it is one of the most popular choices for TTI generation. One may argue that this would limit the possibilities of SynthCLIP to the OpenAI CLIP performance. First, let us highlight how *only the textual encoder of CLIP is used in Stable Diffusion*. CLIP is defined as a pair of encoders mapping text and images to a joint embedding (Radford et al., 2021b). While Stable Diffusion uses a pre-trained CLIP text encoder for prompt interpretation, the CLIP visual encoder is not used anywhere in the pipeline. Also, the usage of CLIP for textual encoding is a design choice in Stable Diffusion, but other popular models such as Imagen (Saharia et al., 2022) rely on textual encoders like T5-XXL (Raffel et al.,

2020) trained on text only, hence training on the text-images real datasets of CLIP is not strictly necessary for the SynthCLIP pipeline.

To prove that we are not limited by the pretrained text encoder performance, we fine-tune the OpenAI CLIP (ViT-B/16), pre-trained on 400 million captioned images, on SynthCI. The finetuning is done on 1M samples, for a single optimization step, in order to avoid loss of performance due to catastrophic forgetting. We report results in Table 8, where we improve over the baseline in four tasks out of five, with $\Delta_{\text{MTL}} = +0.56\%$. Our improved performance suggests that our approach allows us to extend even the capabilities of the CLIP model used in the text-to-image pipeline, by adding more synthetic data to the training.

## F   LIMITATIONS

Here, we discuss limitations. The most evident disadvantage of synthetic CLIP models is the computational effort required to generate the training dataset, which may lead to a high carbon impact. Our generation process currently takes approximately 6.5 days using a 48-A100-80GB GPU cluster, equivalent to 313 GPU days. However, recent advancements in text-to-image and large language models have not only enhanced generation quality but also accelerated inference speeds (Sauer et al., 2023; Kwon et al., 2023). With continual technological advancements, we anticipate a reduction in the time required for generation, leading to more efficient and scalable end-to-end approaches. Future research should include further experimentation with various language models, text-to-image generators, and caption generation prompts to identify optimal configurations for improvement. Moreover, there may be concerns related to copyright issues and memorization in text-to-image diffusion models (Carlini et al., 2023). With the advancements in differential privacy for generative models (Cao et al., 2021), these could be solved in the near future. Finally, we acknowledge that although we train our CLIP on synthetic data only, both the LLM and the TTI have been trained on real data. Nevertheless, it is well-known the capacity of generative networks to create new content *not included* in the dataset (Rombach et al., 2022), factor also proved in our experiments.

## G   ADDITIONAL DETAILS ON $\Delta_{\text{MTL}}$

MTL Vandenhende et al. (2021) is a metric that evaluates the relative performance across multiple tasks by normalizing improvements or degradations with respect to baseline performance. Specifically: For each task $i$, given its baseline performance $b_i$, observed performance $m_i$, and direction of improvement $g_i$ (0 if higher values are better, 1 if lower values are better), the relative performance improvement or degradation is computed as:

$$\Delta_i = (-1)^{g_i} \frac{(m_i - b_i)}{b_i}$$

The final MTL score is the mean of all task-level relative performance scores, expressed as a percentage:

$$\Delta_{\text{MTL}} = \frac{\sum_{i=1}^{N} \Delta_i}{N} \times 100$$

## H   ZERO SHOT PERFORMANCE ON ROBUSTNESS IMAGENET VARIANTS

In an attempt to understand the effective robustness of CLIP models trained on real data, synthetic data, and hybrid-approaches we explore the performance of various models on ImageNetV2, ImageNet-A, ImageNet-R, ImageNet-O, ImageNet-Sketch, and ObjectNet. The results are summarized in Table 9 and show that training on purely synthetic data does not exhibit better performance on robustness datasets. However, the Mixed (hybrid) baseline from Table 7 and the Finetuning (hybrid) baseline from Table 2 (last row) outperform training on real only (CC12M) and synthetic only (SynthCI-30M) datasets. We hypothesize that this is due to the mitigation of distribution shift due to the inclusion of real data, allowing to exploit the superior quality of the synthetic ones, with improved text-image alignment and concept coverage.

Table 9: **Performance on Robustness Datasets.** Where as training on real data only (CC12M) outperforms training on synthetic data only (SynthCI-30M), the hybrid training from Table 7 outperforms both baselines.

| Dataset | ImageNet1K | ImageNetv2 | ImageNet-A | ImageNet-R | ImageNet-O | ImageNet-Sketch | ObjectNet |
|---|---|---|---|---|---|---|---|
| CC12M | 34.7 | 28.8 | 8.32 | 44.4 | 39.9 | 22.9 | 20.3 |
| SynthCI-30M | 30.7 | 27.0 | 7.42 | 30.0 | 28.6 | 11.9 | 18.6 |
| Mixed (from Table 7) | 39.9 | 34.2 | 12.4 | 50.0 | 40.6 | 26.1 | 26.5 |
| Finetuning (Last Row Table 2) | 38.6 | 33.6 | 13.3 | 47.9 | 38.3 | 24.3 | 25.4 |

Table 10: **Comparing 3M Datasets.** We compare the performance of SynthCI-3M a purely synthetic trained baseline to the performance of training a CLIP model on CC3M and LAION-3M. LAION-3M is a random 3M subset from LAION-2B.

| Dataset | ZS | IR | TR | LP | FS |
|---|---|---|---|---|---|
| CC3M | 14.9 | 33.7 | 42.9 | 63.3 | 74.2 |
| LAION-3M | 14.5 | 24.4 | 33.3 | 66.7 | 77.6 |
| SynthCI-3M | 9.5 | 33.9 | 46.0 | 63.7 | 73.8 |

## I  LAION-3M: EXPLORING A 3M SUBSET FROM LAION

To supplement our experiments, we run a baseline on 3M samples taken from LAION-2B Schuhmann et al. (2022). The results are summarized in Table 10 where we find that LAION-3M exhibits better performance when it comes to linear probing (LP) and few shot (FS) results compared to both SynthCI-3M and CC3M. However, due to the noisy nature of LAION captions, this also leads to a big drop in image retrieval (IR) and text retrieval (TR) results where LAION-3M lags by around 10% to both CC3M and SynthCI-3M in both IR and TR. This further showcases the advantages of having aligned captions and images in the SynthCLIP generation.

## J  ERROR COEFFICIENTS

We aim to evaluate how the error decreases by increasing the size of the dataset. To do so, we plot in Figure 8 the error coefficients for both real and synthetic data for the 5 different tasks that we evaluated on. We use a linear regression to plot the error coefficients by fitting on the reported accuracy values across tasks. Interestingly, we notice similar error coefficients across tasks. This is most likely due to the distribution shift between real and synthetic data. As diffusion models advance towards generating more realistic images, we believe this gap will continue to narrow, resulting in greater benefits from synthetic data across various tasks.

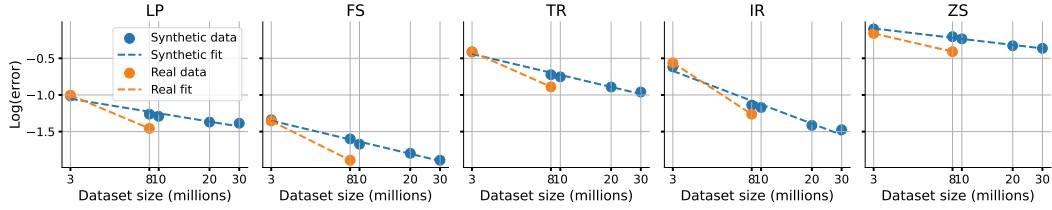

Figure 8: **Error.** Across different tasks, the error tend to decrease in a similar way. This advocates for new techniques compensating the distribution shift to boost all tasks simultaneously.

