# OpenReview forum: "SynthCLIP: Are We Ready for a Fully Synthetic CLIP Training?"
_ICLR.cc/2025/Conference — Submitted to ICLR 2025_

### Official Review · Reviewer_i2N9 · 2024-10-30

**Soundness:** 2
**Presentation:** 3
**Contribution:** 1
**Rating:** 3
**Confidence:** 4

**Summary:**

The paper focuses on training a CLIP-based model using solely synthetic image-text pairs and study the effects of doing so.
SynthCLIP relies on the understanding that curating real multimodal data at scale comes with a cost of quality and alignment between images and their description. To this end, the authors propose to harness the advancement of Text-To-Image models and LLMs to generate a 30M purely synthetic dataset of image-caption pairs. Such a process enables to easily control the distribution of the data and to generate datasets at any scale with no human in the loop.
The authors study the effects of training CLIP with their propose dataset in a various of benchmarks including both vision and vision language tasks and compare it to training with real data.

**Strengths:**

+ The paper is well written and easy to follow.
+ Utilizing a purely synthetic dataset enables to control the data distribution and to collect data and any scale, without requiring human intervention.
+ Interesting and insightful ablation studies

**Weaknesses:**

- Empirical demonstration of the motivation - the authors claim that in real datasets, increasing their scale comes with a cost of quality and the proposed approach mitigates this and enables collecting quality data in any scale. However, empirically proving this requires comparing the performance against much larger real datasets than CC12M. Outperforming a model trained on CC12M with a 30M dataset doesn't showcase the advantage in quality of the synthetic data. Moreover, the model trained with a10M synthetic data has a worse performance compared to the model trained with a similar amount of real data. Thus, I am afraid that the main claim of the paper was not empirically demonstrated.

- Novelty - The proposed framework is based on existing models and approaches. Unfortunately, constructing the concept bank, which could have been an interesting place for novelty, is taken from an existing work.

- The necessity of "human intervention" - real datasets mainly scraped from the internet and the caption is achieved from the alt-text in the html. Thus, curating large datasets is done automatically. Indeed, such datasets are often noisy and there are various methods for filtering them (for example, by a CLIP-score threshold) or bootstrapping [1]. These methods enable collecting huge scale dataset without requiring human intervention. While relying on alt-text for caption often lead to short and oversimplified captions, there are many works that tackle this by proposing automatic recaptioning [2,3]. Thus, one can utilize real images oriented dataset with high quality without human intervention.

- Fairness of comparison in the experimental section - The authors have stated that the training is done for a fixed number of epochs with a fixed batch size for datasets of different size. From my understanding, this leads to a different number of training iterations for datasets at a different scale, impairing the validity of the comparison.

[1] Blip: Bootstrapping language-image pre-training for unified vision-language understanding and generation.
[2] Improving clip training with language rewrites.
[3] FuseCap: Leveraging Large Language Models for Enriched Fused Image Captions

**Questions:**

- In lines 158-159 the authors state that the captioned are oriented for a single object. How would it effect the performance on tasks that require low-level details understanding? There are many works that try to train on detailed captions to incorporate such an understanding.

- TTI models are currently not good at generating text within images and in understanding relationships between objects. Wouldn't training on generated images result in a model with limited capabilities in such areas?

- Given figure 4, why would we need to generate images and not recaption ones that we can obtain easily by crawling the internet?

---

> ### Author Response · Authors · 2024-11-27
> **Response to Reviewer [1/2]**
>
> Thank you for recognizing the clarity of our writing and the advantages of using a synthetic dataset for controlled and scalable data collection. We also appreciate your acknowledgment of our ablation studies as insightful and impactful.
>
> **Main Claim is Not Demonstrated:**
>
> We believe there is a misunderstanding. The main claim of the paper is not to say that synthetic data is superior to real data in all aspects. The paper tries to understand the strengths and limitations of using a fully synthetic pipeline for training CLIP models. We show that on the same scale synthetic data is unable to outperform real data, but upon scaling the gap is reduced and models trained on purely synthetic data, yet with a lower sample efficiency, can outperform CC12M training. This is a novel contribution, since it is unclear how synthetic data representing a large set of concepts would perform in training, and how much they will underperform compared to real data. We also show two hybrid approaches: (1) Fine Tuning a model trained on pure synthetic data on a few real samples, and (2) Training from scratch on a hybrid dataset (synthetic + real samples). These approaches also quantify how much pre-training on synthetic data will differ from joint training in real and synthetic data.
>
> **Novelty:**
>
> We respectfully but strongly disagree. First, let us highlight that though our concept bank is adopted from MetaCLIP, using 500 thousand concepts for generation allows us to draw significant insights on the impact of distributions in training. Existing works [1,2,3] all assume knowledge of the evaluation benchmarks and heavily bias the generated data to align with the downstream evaluation classes. We also studied various selections of the concept bank such as selecting a random subset of concepts or limiting the concept bank to be the concepts seen in CC3M dataset (Table 4). Our results show that limiting the generation to CC3M concepts (40K concepts) gives you a boost in performance over generation for all 500K concepts of MetaCLIP. On the contrary, generating and training on a random 40K subset (i.e same scale as CC3M concepts but random) leads to lower results compared to 500K concepts of MetaCLIP. Those insights are novel and have not been explored previously in other papers.  We also note the following works, published in top-tier conferences, that did not introduce new components but relied on smartly connecting them together or prompting [4,5]. We kindly ask you to reconsider your assessment.
>
> **Human Intervention:**
>
> While naive data curation pipelines may still lead to usable datasets, these strategies are highly suboptimal and might still violate regulatory policies and might contain prohibited content such as “child abuse” which was found in LAION-5B dataset [6]. Let us also stress that simply recaptioning while ignoring distribution balancing may not fully solve the curation problem, as there are many works relying on advanced data curation pipelines that involve either computational or engineering complexity [7,8,9,10]. Ultimately, we agree to specify this in the paper, but we ask you to reconsider the importance of the property of synthetic data as regards generation of curated data at scale.
>
>
> **Fairness in Comparisons:**
>
> This is a common setup in self-supervised learning, and many considerably cited approaches just focus on absolute performance disregarding computational costs [11,12]. Moreover, we observed that CC3M and CC12M datasets reached its highest performance well before completing 40 epochs, with CC12M plateauing at epoch 29. In contrast, SynthCI-30M continued to show improvement up to the 40th epoch. To further investigate this, we conducted an additional experiment using SynthCI-3M, training it for 100 epochs to match the total sample exposure of SynthCI-7.5M trained for 40 epochs. The results showed early performance saturation, with the highest zero-shot accuracy (9.8%) occurring at epoch 24. This experiment suggests that extending training duration on a smaller dataset may not yield any benefits, proving the validity of our comparison.

---

> > ### Author Response · Authors · 2024-11-27
> > **Response to Reviewer [2/2]**
> >
> > **Single Objects in Captions:**
> >
> > Even though we prompt the LLM to generate a caption around a single concept, we observe that the generated captions have much wider coverage of concepts compared to real captions. This is due to the natural emergence of multiple concepts in the generated captions, and it justifies our choice to use balanced sampling. In Table-5 we show that even the smallest SynthCI-3M dataset contains significantly more concepts than the larger real CC12M dataset.
> >
> > **Understanding Capabilities of TTI:**
> >
> > In our experiments we use StableDiffusion v1.5 as per previous literature (StableRep). However, our pipeline is not limited to specific TTI models and it can be swapped with more recent TTI models that are of much higher fidelity. Current TTI models use much more sophisticated techniques to improve compositionality compared to StableDiffusion v1.5. Regardless of this, our findings are independent from the used model.
> >
> >
> > **Recaptioning Images from the Web:**
> >
> > It is important to note that we are not advocating the use of purely synthetic data in all cases, but rather to study the effects of pretraining on synthetic data in cases in which data alignment, scalability without human intervention, and safety of generated data are crucial. We are open to further clarify this.
> >
> >
> >
> > **References:**
> >
> > [1] Learning Vision from Models Rivals Learning Vision from Data (CVPR 2024)
> >
> > [2] Scaling Laws of Synthetic Images for Model Training ... for Now (CVPR 2024)
> >
> > [3] Is synthetic data from generative models ready for image recognition?
> >
> > [4] Improving CLIP Training with Language Rewrites (NeurIPS 2023)
> >
> > [5] VeCLIP: Improving CLIP Training via Visual-enriched Captions (ECCV 2024)
> >
> > [6] https://www.telegraph.co.uk/business/2023/12/20/fears-ai-trained-child-abuse-images-thousands-discovered/
> >
> > [7] Scaling Laws for Data Filtering-- Data Curation cannot be Compute Agnostic (CVPR 2024)
> >
> > [8] DINOv2: Learning Robust Visual Features without Supervision (TMLR 2024)
> >
> > [9] The Role of Data Curation in Image Captioning (EACL 2024)
> >
> > [10] CiT: Curation in Training for Effective Vision-Language Data (ICCV 2024)
> >
> > [11] Self-supervised Pretraining of Visual Features in the Wild
> >
> > [12] DINOv2: Learning Robust Visual Features without Supervision (TMLR)

---

### Official Review · Reviewer_xQmA · 2024-11-02

**Soundness:** 3
**Presentation:** 3
**Contribution:** 3
**Rating:** 6
**Confidence:** 4

**Summary:**

The work proposes a synthetic training protocol for CLIP models extending prior work by creating both generated captions and generated images. In the process, the method demonstrates superior performance to common small scale image-text such as CC12M by curating a dataset of 30M synthetic examples. To better understand how different elements of the pipeline affect performance, they also ablate different choices of language models, differences caused by synthetic data sources, and how concept distribution impacts performance.

**Strengths:**

Each element of the synthetic data pipeline is soundly constructed, and follows community norms for composition. Furthermore, the work details a lot of the smaller design choices (e.g. LLM, prompt, and concept distribution) that contribute to the end to end system. The approach itself is noteworthy as it represents a full departure to synthetic data whereas most existing approaches require one of the modalities to be pre-existing.

Model benchmarks are extensive and representative of image-text model capabilities. They show both good diversity in dataset and task.

Some of the most interesting findings come from the paper's ablations. Figure 4a demonstrates the delta in performance that can be attributed to each different synthetic modality. In addition, it shows a potential failure mode of ignored generation commands and how they may be addressed. Additionally, the study on the concept bank is quite interesting providing support for the hypothesis that some of the difference in performance between natural and synthetic data is from the underlying conceptual distribution not a failure in quality. The experiments towards the mitigation of long-tail effects suggest an interesting direction for improving unseen or undersampled concepts in real world training.

**Weaknesses:**

One concern with this work is that there is not sufficient evidence that the method might scale. Certain CLIP pretraining augmentations, like M3AE for example, have been shown to work at small scales, but yield no major benefit at larger scales [1]. Understanding that training frontier CLIP models is prohibitively expensive due to batch size needs, it’s sensible that this data is not available but worth keeping in mind.

The most immediate notice is the difference in data efficiency between real and synthetically drawn samples. Combined with the above, for practical applications it is not quite clear when one would adopt this method as it strictly leads to longer training runs and real training data is abundant (LAION-5B [2] and DataComp [3]). The method would benefit from further analysis into what is causing the reduction in performance, though some beginning analysis is done with respect to the concept distribution the gap is still left largely unexplained.

[1] Weers et al. 2023 "Masked Autoencoding Does Not Help Natural Language Supervision at Scale"
[2] Schuhmann et al. 2022 "LAION-5B: An open large-scale dataset for training next generation image-text models"
[3] Gadre et al. 2023 "DataComp: In search of the next generation of multimodal datasets"

**Questions:**

1. It isn’t quite clear how MTL is calculated. What is it averaging over?
2. In an effort to understand differences in synthetic distributions versus natural distributions, how does performance change when using a CC3M concept distribution sample equalized to the real CC3M similar to Table 4? Another experiment to get at some of this would be taking CC3M, for each image prompting the model for a single “concept” then creating a caption and image using the proposed pipeline.
3. To understand the differences in scaling, it would be helpful to know the coefficients of the error with respect to dataset size on a log scale. How do natural and synthetic data coefficients compare?
4. This experiment is less pertinent than the above, but with results on improving the long tail distribution there might be interesting robustness properties as a result of concept representation. How does the real versus natural data compare on effective robustness in a framework like that of [1]?

Overall, the work is well presented but would benefit from a coverage of the first three points, and less importantly the fourth, to round out its presentation. I’d be happy to raise my score if the above are addressed.

[1] Nguyen et al. 2023 "Quality Not Quantity: On the Interaction between Dataset Design and Robustness of CLIP"

---

> ### Author Response · Authors · 2024-11-28
> **Response to Reviewer [1/2]**
>
> Thank you for your thoughtful and detailed review. We deeply appreciate your recognition of our synthetic data pipeline design and the comprehensive benchmarks and ablations we conducted. Your insights, especially regarding our conceptual distribution and long-tail experiments, highlight some of the key contributions we aimed to achieve in our analysis.
>
> **Scalability of the Method:**
>
> We acknowledge that the performance at a larger scale is not fully known. While 30M samples might not be at the scale of existing large-scale real datasets, it does allow for understanding the strengths and limitations of synthetic data for training CLIP models. Through that scale of data, we were able to provide interesting insights such as understanding the effect of the concepts distribution (Table 3), the importance of various modalities (Figure 4), and the effect of choice of language models (Table 3). Our paper serves as a stepping stone for defining possible good practices in synthetic training of CLIP models. We do not claim to solve the problem entirely, however we hope this project, and all the assets (code, data and models) that will be open-sourced to help the research community. Even though M3AE explored a relatively small scale for data, their impact on the community is reflected by the citations on that work.
>
> **Delta MTL Calculation:**
>
> MTL is a metric that evaluates the relative performance across multiple tasks by normalizing improvements or degradations with respect to baseline performance. Specifically:
> For each task $i$, given its baseline performance $b_i$, observed performance $m_i$, and direction of improvement $g_i$ ($0$ if higher values are better, $1$ if lower values are better), the relative performance improvement or degradation is computed as:
>
> $$\Delta_i = (-1)^{g_i} \frac{(m_i - b_i)}{b_i}$$
>
> The final MTL score is the mean of all task-level relative performance scores, expressed as a percentage:
> $$
> MTL = \frac{\sum_{i=1}^{N} \Delta_i}{N} \times 100
> $$
>
> We added this to the revised manuscript.
>
> **Training with CC3M Concept Distribution:**
>
> While this is an interesting experiment, it would be hard to implement due to the following reasons: (1) Given a caption from the CC3M, deciding on which concept of the many concepts that appear in the caption is the one we generate the synthetic caption and image for is non trivial. (2) Assuming we select the main subject of the caption to the be concept, which will lead to loss of certain concepts, we cannot control which concepts the language model will decide to add to the generated caption, hence the control over the distribution is very challenging.
>
> Instead we explored the impact of concept distribution by comparing performance when training SynthCLIP on CC3M-specific concepts ( $C_{CC3M}$) versus random subsets ($C_{\text{rand}}$). $C_{CC3M}$ was created by identifying approximately 40,000 concepts in our bank that overlap with CC3M captions, while $C_{\text{rand}}$ contained 40,000 randomly chosen concepts. Training on $C_{CC3M}$ led to better performance in tasks like text retrieval ($+3.9\%$) and linear probing ($+1.6\%$), likely due to alignment with downstream datasets, reflecting a distribution bias in CC3M toward commonly evaluated tasks. Conversely,  $C_{\text{rand}}$ underperformed across benchmarks, showing the importance of concept relevance.
>
>
> **Log-Scale Error Plots:**
>
> Thanks for the interesting suggestion, which led to interesting findings. We performed the experiment and included it into the supplementary with a new plot. Please refer to Figure 8. Overall, we observe that coefficients are similar across tasks for real and synthetic data, respectively. We believe that the main cause of such behavior is the distribution shift, that is impacting the learned representation equally for each task. With future developments of diffusion models allowing for compensating such distribution gaps, it is realistic to assume that the tasks analyzed would be improved.
>
>
> **Long-tail Concepts:**
>
> Thanks again for the interesting experiment proposed. We followed your suggestion and evaluated the effective robustness properties of SynthCLIP on different datasets similarly to [1], as also suggested by Reviewer c8nx. We present results in Table 9 in the supplementary material. Overall, we do not observe robustness properties due to synthetic data usage. However, mixing real with synthetic data in a similar fashion to those explored in Table 7 allows us to get best performance. We hypothesize that this is due to the simultaneous compensation of the distribution shift (due to the inclusion of real data) and the improved quality of the representations (due to the quality of synthetic data and coverage of concepts).

---

> ### Author Response · Authors · 2024-11-28
> **Response to Reviewer [2/2]**
>
> **Real vs Synthetic Data:**
>
> Although synthetic data falls short of real data in transfer effectiveness on real-world datasets, we emphasize that uncurated web datasets like LAION and DataComp could face stricter regulatory scrutiny compared to synthetic data, given the growing attention from regulatory bodies. By examining the characteristics of training on synthetic data, we aim to lay the groundwork for developing new foundation models that are entirely free from real data, and thus avoid such regulatory challenges. The performance gap between synthetic and real data is well-documented and widely acknowledged, with extensive literature exploring the distribution shift between the two. While investigating the underlying causes of this gap is an interesting direction, it remains an unresolved issue despite years of research [2] and falls outside the scope of SynthCLIP. Instead, our approach acknowledges the effects of distribution shift and focuses on studying the unique properties of networks trained exclusively on synthetic data.
>
> **References:**
>
> [1] Quality Not Quantity: On the Interaction between Dataset Design and Robustness of CLIP, NeurIPS 2022
>
> [2] Visda: A synthetic-to-real benchmark for visual domain adaptation, CVPRw 2018

---

### Official Review · Reviewer_c8nx · 2024-11-04

**Soundness:** 3
**Presentation:** 2
**Contribution:** 2
**Rating:** 5
**Confidence:** 4

**Summary:**

It's interesting to see the emerging trend of training CLIP models using synthetic data. This work introduces SynthCLIP, a CLIP model training on synthetic data comprising both synthetic captions and images. The paper not only proposed the pipeline for creating synthetic data but also release SynthCI-30M, a comprehensive dataset housing 30 million captioned images generated entirely synthetically. This work unveiled the potential of leveraging synthetic data to enhance CLIP model training.

**Strengths:**

1. The research detailed the pipeline for entirely synthetic creation of image-text pairs and introduced the dataset SynthCI-30M.

2. A comprehensive set of experiments was conducted to elucidate the efficiency and practical value of synthetic data, demonstrating its scalability and effectiveness.

**Weaknesses:**

1. The zero-shot image classification results lack depth to gauge effectiveness comprehensively.

2. The experiments solely compared with CC3M and CC12M. How would results differ if a subset of LAION-400M is employed instead?

**Questions:**

1. I'm curious about the performance on ImageNet variants like ImageNetV2, ImageNet-A, ImageNet-R, and ObjectNet.

2. Will the performance of MLLM understanding tasks be enhanced by employing CLIP trained on SynthCI-30M?

---

> ### Author Response · Authors · 2024-11-27
> **Response to Reviewer**
>
> Thank you for acknowledging the significance of our pipeline and dataset, as well as the rigor and depth of our experiments. We greatly appreciate your thoughtful review and encouraging feedback.
>
> **Depth of Zero-Shot Image Classification Results and ImageNet Variants:**
>
> We appreciate the feedback and agree that a more comprehensive evaluation would strengthen our findings. Before presenting additional experiments, let us highlight that our evaluation goes way beyond zero-shot image classification, including also linear probing, few-shot learning, text retrieval, and image retrieval tasks across multiple datasets (refer to Table 1a and 1b). We will highlight the fact that higher zero-shot accuracy correlates with improved performance across tasks.
>
>
> Following your suggestion, we have conducted further zero-shot evaluations on ImageNet variants such as ImageNetV2, ImageNet-A, ImageNet-R, ImageNet-O, ImageNet-Sketch, and ObjectNet. We will include these results in the revised manuscript.
> | Dataset                      | ImageNet1K | ImageNetv2 | ImageNet-A | ImageNet-R | ImageNet-O | ImageNet-Sketch | ObjectNet |
> |------------------------------|------------|------------|------------|------------|------------|-----------------|-----------|
> | **SynthCI-30M**              | 30.7       | 27.0       | 7.42       | 30.0       | 28.6       | 11.9            | 18.6      |
> | **CC12M**                    | 34.7       | 28.8       | 8.32       | 44.4       | 39.9       | 22.9            | 20.3      |
>
> As shown in the previous table, the performance gap is similar to the one in ImageNet, suggesting that the distribution gap is the most influential source of performance degradation. To further provide intuitions on settings involving both real and synthetic sata, we evaluated in two additional setups: 1) the **finetuning** setup following the settings of Table 2, and the **mixed** setup of Table 7 in the appendix. Results are below:
> | Dataset         | ImageNet1K | ImageNetv2 | ImageNet-A | ImageNet-R | ImageNet-O | ImageNet-Sketch | ObjectNet |
> |-----------------|------------|------------|------------|------------|------------|-----------------|-----------|
> | **Finetuning** | 38.3       | 33.6       | 13.3       | 47.9       | 38.3       | 24.3            | 25.4      |
> | **Mixed**      | 39.9       | 34.2       | 12.4       | 50.0       | 40.6       | 26.1            | 26.5      |
>
>
> As visible, performance still follow the ones reported in the main paper: mixing data performs best, but finetuning synthetic-pretrained representation extractors already allows for a major boost in performance.
>
> **Comparison with LAION-400M:**
>
> Thanks for the interesting suggestion. We performed the experiment and trained on a random 3M images subset sampled from LAION-400M. Here, we report the results.
>
> | Dataset     | ZS   | IR   | TR   | LP   | FS   |
> |-------------|------|------|------|------|------|
> | LAION      | 14.5 | 24.4 | 33.3 | 66.7 | 77.6 |
> | CC3M       | 14.9 | 33.7 | 42.9 | 63.3 | 74.2 |
> | SynthCI-3M | 9.5  | 33.9 | 46.0 | 63.7 | 73.8 |
>
> In particular, let us highlight the significant drop in IR and TR due to the usage of LAION. We attribute this to the non-descriptive captions, lacking the data curation of CC3M. This is further proof of the benefits of synthetic data in this case, considering that we allow for improved performance in IR and TR without human data curation as in CC3M.
>
> **Performance on ImageNet Variants and MLLM Understanding Tasks:**
>
> While MLLM requires more sophisticated textual encoders, and as such training such a model goes beyond the scale of data investigated in our paper, we believe there is a reasonable expectation that the superior alignment between text and images of synthetic data would improve reasoning performance. Throughout the paper we reported results on Image Retrieval (IR) and Text Retrieval (TR) which require knowledge of both text and image, as MLLM reasoning tasks. Our experiments showed that training on synthetic data at scale exhibits the highest IR and TR accuracies (for example Table 1-b SynthCLIP with 30M samples achieve 61.7% on IR and 77.1% on TR compared to 58.9% and 71.7% achieved by CC12M training).
> Additionally, Table 8 shows the results of fine tuning OpenAI CLIP on our generated synthetic data and we find that the model performance increases on both IR and TR highlighting the effectiveness of synthetic data on vision-language understanding of the model.

---

### Official Review · Reviewer_xUMB · 2024-11-07

**Soundness:** 2
**Presentation:** 2
**Contribution:** 2
**Rating:** 5
**Confidence:** 4

**Summary:**

The paper explores the performance of CLIP-style models trained on purely synthetic image-caption pairs (called SynthCLIP) generated by modern text-to-image diffusion models and LLMs. It studies the scaling trends of such models and also provides a dataset of 30 million captioned images.

**Strengths:**

- The paper provides a dataset of 30 million captioned images from diverse set of concepts.
- As the concepts are fixed, one can gather a subset of them to train new models with less concerns about NSFW contents leaking into the training data compared to using real images.
- The models trained with similar dataset scales using synthetic captioned images show similar performance on down-stream tasks.

**Weaknesses:**

- The main weakness of the paper in my opinion is that the paper is not well-motivated. The introduction section does not provide convincing answers to the questions like "why should we use purely synthetic image-caption datasets? why not a hybrid approach? why is the problem significant?"

- Although controlling the concepts that are present in the dataset can be useful, there is no guarantee that the generated images for each concept are 1) faithful to the content and 2) do not contain NSFW content:
1) faithful to content: The described workflow only filters the captions, not the generated images. It is likely that the generated images contain noisy unrelated images. No workarounds in this regard has been proposed in the paper.

2) NSFW content: The latter may happen because the training of models like Stable diffusion have been on unfiltered datasets like LAION. It is possible that some NSFW contents appear with some concepts in these datasets frequently, resulting in generation of NSFW contents inadvertently.

- Despite that the paper argues that one can use synthetic images from tail classes to augment the real datasets, I think it is not straightforward to do so. Although the idea seems sound, the Stable Diffusion (SD) model has been trained on real images that have the same long-tailed classes. Therefore, the performance of SD on these classes will not be satisfactory.

**Questions:**

- I suggest that the authors improve the introduction by explaining the motivations and use-cases of employing a purely synthetic dataset to train CLIP models.

---

> ### Author Response · Authors · 2024-11-27
> **Response to Reviewer [1/2]**
>
> Thank you for recognizing the value of our dataset and its potential to enable safer and effective model training, as well as highlighting the comparable downstream performance of models trained on synthetic captions. Your thoughtful feedback is greatly appreciated.
>
> **Motivation for Using Purely Synthetic Image-Caption Datasets:**
>
> We appreciate this observation and acknowledge the need to clarify our motivations. Our work aims to explore the potential and limitations of synthetic data for pre-training CLIP-like models, rather than advocating exclusively for synthetic data. We think that training on synthetic data will be increasingly important in the next future. Indeed, while collecting data on the web is arguably easy, curating such data is a cumbersome and cost-inefficient operation. Synthetic data, conversely, allow for an automatic data curation, and for having control over the content generated. This not only allows us to benefit from strong composition capabilities, allowing it to hallucinate objects difficult to collect in real life (e.g., an elephant on the moon), but also enable further control over copyrighted and safe content. We believe this will also increase in the future due to the efforts in differentially private [1] and safe [2] diffusion models.
>
> While our experiments demonstrate that hybrid approaches (e.g., fine-tuning on real images after pretraining on synthetic data) outperform purely synthetic models, our primary goal is to understand the standalone capabilities and limitations of synthetic data due to the aforementioned advantages. Let us highlight that in the main paper we included a hybrid approach on small curated real data (Table 2), which is a realistic scenario in presence of a large-scale synthetic dataset, automatically curated.
>
> We updated the abstract of the paper to clarify this.
>
>
> **Faithfulness to Content and NSFW Content in Generated Images:**
>
> Although we do not provide guarantees on the generated content, these are open problems in text-to-image generators, with significant efforts in the state-of-the-art to achieve faithful [3,4,5] and safe [2,6] outputs. Since our formulation is general, with further research on the topic, these issues will be solved by the diffusion model used for generation. However, we did our best effort to quantify the effects of misalignment between captions and images, and NSFW outputs. Both impact marginally our results. In particular:
>
> - **Faithfulness to Content:** We conducted experiments where we recaptioned generated images (Table 4a). This process improved performance across most tasks, demonstrating that enhancing caption post-generation can quality can mitigate content unfaithfulness. However, we managed to scale the training even with original captions.
> - **NSFW Content:** Our dataset analysis revealed that approximately 3.15% of the MetaCLIP 500k concepts are NSFW concepts. To eliminate NSFW generations, we implemented a filtering mechanism to exclude NSFW concepts from our concept bank, thereby reducing the incidence of NSFW content in generated images. This is presented in Section 6. Let us also highlight the uncurated LAION-5B approximately 3% NSFW concepts [7], also containing illegal child abuse images regardless of security checks [8]. This emphasizes the challenge of collecting safe real data at scale, and advocates for the advantages of our synthetic generation. To further support our argument, we applied an NSFW detector [9] to the SynthCI-30M images after filtering, revealing only 0.005% NSFW content. This shows that the impact of safety degeneration is marginal.
>
>
> **Augmenting Real Datasets with Synthetic Images from Tail Classes:**
>
> We appreciate the reviewer’s feedback and acknowledge the limitations of generative models like Stable Diffusion in representing tail classes. As detailed in the discussion section, our experiments demonstrate that augmenting datasets with synthetic images improves downstream task performance, even for tail classes. Specifically, we observed significant improvements in zero-shot classification accuracy for 10 tail classes: 44.18% accuracy for CLIP versus 60.04% accuracy for SynthCLIP, with 150 samples per class. We attribute this behavior to the synthesis of parts or patterns typical of those classes, that may be rendered realistically even though the capabilities of Stable Diffusion on such elements are limited.
> These results illustrate that synthetic data can enhance performance by increasing diversity and coverage, especially in challenging long-tail distributions. While we recognize existing limitations in generative model performance for tail classes, these findings suggest synthetic augmentation is a promising approach.

---

> > ### Author Response · Authors · 2024-11-27
> > **Response to Reviewer [2/2]**
> >
> > **References:**
> >
> > [1] Differentially Private Diffusion Models, TMLR 2023
> >
> > [2] Mitigating Inappropriate Degeneration in Diffusion Models, CVPR 2023
> >
> > [3] Imagen Editor and EditBench: Advancing and Evaluating Text-Guided Image Inpainting, CVPR 2023
> >
> > [4] Conform: Contrast is all you need for high-fidelity text-to-image diffusion models, CVPR 2024
> >
> > [5] Compositional Visual Generation with Composable Diffusion Models, ECCV 2022
> >
> > [6] [6] Erasing concepts from diffusion models, ICCV 2023
> >
> > [7] LAION-5B, NeurIPS 2022
> >
> > [8] https://www.telegraph.co.uk/business/2023/12/20/fears-ai-trained-child-abuse-images-thousands-discovered/
> >
> > [9] Can Machines Help Us Answering Question 16 in Datasheets, and In Turn Reflecting on Inappropriate Content? Facct 2022

---

### Meta-Review · Area_Chair_GM1L · 2024-12-19

**Metareview:**

This paper discusses using pure synthetic data for training CLIP models, and shows the performance, scaling property, and analysis compared with real datasets. The paper also introduces a new SynthCI-30M with captions on 30 million images. However, the paper still lacks enough evidence on several key points such as the quality of content of generation images, not enough large-scale experiments and ablations, etc. The motivation of the work also needs to be justified. Therefore, based on the reviews, I recommend rejection of the paper.

**Additional Comments On Reviewer Discussion:**

Reviewer c8nx asked for more evaluations, which were added by the authors during the rebuttal. Reviewer xUMB asked for a better statement of motivation and the quality of generated images in terms of faithfulness and NSFW content. The authors re-stated that the motivation is focused on the discussion of synthetic data and listed reasons for doing this compared to using real data. For the content quality, the authors mentioned there are some discussions and analyses on those already. Reviewer xQmA asked about scaling of the data and together with reviewer xUMB, they asked about the long-tail classes by image generation models. The authors added some discussions on this in the supplementary. Reviewer i2N9 is concerned with motivation, novelty, human intervention, and fairness of comparison. The authors replied with references and data points in the submission, but seems not all the points are fully addressed.

---

### Decision · Program_Chairs · 2025-01-22

Reject